# Effect of Ammonia Addition on the Growth of an AlO(OH) Film during Steam Coating Process

Naotaka Itano, So Yoon Lee and Ai Serizawa *

Department of Materials Science and Engineering, Shibaura Institute of Technology, College of Engineering, Tokyo 108-8548, Japan; mb21007@shibaura-it.ac.jp (N.I.); soyoon@shibaura-it.ac.jp (S.Y.L.)
* Correspondence: serizawa@shibaura-it.ac.jp

**Abstract:** Al alloys possess excellent physical and mechanical properties, such as low density, high specific strength, and good ductility. However, their low corrosion resistance limits their use in corrosive environments. The steam coating process has attracted considerable attention as a new coating technology that can improve the corrosion resistance of Al alloys. This surface treatment technology uses steam to form a corrosion-resistant film on Al alloys. However, a decrease in the processing time, which can result in a lower cost, is needed for the practical application of the steam coating process. In this study, an Al-Mg-Si alloy is used as the base material, and ammonia is added to the steam source to increase the film formation rate. By adding ammonia (0.5 mol/L) to the steam source, the rate constant, K, for film formation increases 1.82 times compared to that of the pure-water-only treatment. Field emission scanning electron micrographs of the film surface confirms that the crystal morphologies of the crystals change from rectangular to parallelepiped shape with increasing process time by ammonia addition. Furthermore, X-ray diffraction patterns show that AlO(OH) crystals are successfully synthesized without byproducts, even when ammonia is added.

**Keywords:** steam coating process; aluminum alloy; anticorrosive film; surface treatment





## 1. Introduction

In recent years, lightweight materials, including aluminum, magnesium, plastics, and composite materials, have been attracting increasing attention for reducing vehicle weight and improving fuel efficiency [1]. Among these materials, Al is an ideal structural material due to its superior mechanical and physical properties. As a result, Al alloys are used as structural materials in automobiles, high-speed rail vehicles, and aircraft applications [2].

Al-Mg-Si alloys (AA6000 series Al alloys) are lightweight heat-treatable alloys with medium strength, which are used in a wide range of applications, such as transportation [3,4], construction [5], and high-voltage power transmission [6]. In addition, Al-Mg-Si alloys are used to manufacture lightweight automotive panels, as they exhibit an excellent precipitation hardening reaction at the same temperature (approximately 170–180 °C) used as a paint baking process in the manufacturing process of vehicles [7–12]. However, Al-Mg-Si alloys have low corrosion resistance. To overcome this deficiency, an improvement in corrosion resistance via the application of coatings is required to enable their use as structural materials.

It is practical to form an oxide film on the surface [13] in order to improve the corrosion resistance of Al. Oxide films formed on Al surfaces can be divided into natural oxide films and those produced by the artificial treatment. Natural oxide films are formed through contact with air [14] and have a thickness of 1.6–2.5 nm [15]. Because natural oxide films are very thin and have non-uniform structures, they do not prevent the corrosion of alloys [13].

Various surface treatment techniques have been developed and implemented to improve the corrosion resistance of alloys. Specifically, surface treatment techniques, such as anodic oxidation [16–19], electroplating [20,21], conversion treatment [22], and sol gel

method [23,24] have been industrially used for different kinds of Al alloys. However, in these surface treatment processes, the waste liquid generated needs to be treated before disposal, and there is a risk of environmental pollution caused by heavy metals [25]. In recent years, global environmental issues have become a major concern, and efforts are being made to reduce the use of environmentally hazardous substances in automobile production [26]. In particular, the Sustainable Development Goals (SDGs), which are international goals aimed at creating a sustainable and better world by 2030, are being actively addressed worldwide, and 4 of the 17 SDGs are related to environmental conservation [27]. Therefore, there is a strong demand for environmentally friendly means of treating Al alloys.

Based on these factors, this study was focused on the steam coating process, which uses steam generated in a pressure vessel (autoclave) to form a dense anticorrosive film on the metal surface [28,29]. The advantage of this surface technology is that it is environmentally friendly, as only water is used in the process. However, a reduction in the processing time resulting in a lower production cost is needed for practical application. In this process, an AlO(OH) film is formed from the reaction of water vapor and an Al, but the rate of this chemical reaction can vary depending on various factors, such as treatment temperature and pressure, the reactant concentration, and the presence of catalysts or inhibitors [30]. It has been reported that adding ammonia during the boiling step in the steam process reduces the processing time [31]. Ammonia was chosen because it is a weak base and has a low environmental impact compared to strong bases, such as sodium hydroxide, thus addressing some of the environmental issues. In addition, in recent years, ammonia technology has been developed to lower $CO_2$ emissions in the manufacturing process, and waste-water treatment technology for ammonia has been developed [32,33]. In this study, ammonia was added to the steam source to increase the film formation rate, which was demonstrated by observing the surface morphology and cross-section of the film.

## 2. Materials and Methods

An Al-Mg-Si alloy substrate with dimensions of 20 mm × 20 mm and 10 mm × 10 mm was used. The alloy was supplied as a 1 mm cold-rolled sheet, and its chemical composition is listed in Table 1.

**Table 1.** Chemical composition of the alloy used in the present study (wt%).

| Mg | Si | Cu | Mn | Fe | Cr | Zn | Ti | Al |
|------|------|------|------|------|------|------|------|------|
| 0.59 | 0.96 | 0.01 | 0.05 | 0.18 | 0.04 | 0.01 | 0.02 | Bal. |

The substrate was subjected to solution treatment at 560 °C for 30 min and quenching into iced water at 0 °C for 1 min. After cleaning with ethanol, it was subjected to the steam coating process at 200 °C for 8, 16, and 24 h. Ultrapure water and 28% ammonia solution (Wako Pure Chemicals Co.) were used as the steam source. The steam source was prepared as 0.1, 0.3, and 0.5 mol/L of ammonia solution using ultrapure water. The total volume of solution was 10 mL, and the pH was measured in the ammonia added mixture before and after the treatment. For specimen preparation, the temperature for the steam coating process was set to 200 °C throughout this study. Ammonia-treated specimens were prepared by adding 0.1, 0.3, and 0.5 mol/L of ammonia solution into the autoclave. Ultrapure-water-treated specimens were also prepared by adding 10 mL of ultrapure water as a control. The ultrapure-water-treated specimens were labeled as pure water and ammonia-treated specimens as 0.1, 0.3, and 0.5 mol/L, corresponding to the concentration. In this study, two specimens were prepared for each condition in order to confirm the duplicability, and surface and cross-sectional observations and evaluations were carried out.

The surface and cross-sectional morphologies of the anticorrosive films on the Al-Mg-Si alloy substrates were observed using a field emission scanning electron microscopy (FE-SEM, JSM-7610F, JEOL Ltd., Tokyo, Japan) operating at 6 kV. The crystal phase of

the obtained film was identified using glancing-angle X-ray diffraction (XRD, Smart Lab, Rigaku, Tokyo, Japan) at a glancing angle of 1° with Cu Kα radiation (40 kV, 30 mA) within a range of 5–90° and at a scanning rate of 2θ = 4°min$^{-1}$. A cross-section polisher (CP, IB-09010CP, JOEL Ltd., Tokyo, Japan) was used for cross-section creation using Ar$^{3+}$ ions accelerated at 6.0 kV [34].

## 3. Results

### 3.1. pH Measurement

The results of pH measurements before and after the treatment are shown in Figure 1. The standard deviation of pH measurements before and after treatment at each time are shown in Table 2. The pH values decreased after the steam coating process for all conditions. The overall chemical reaction which determines pH values can be expressed as

$$2Al^{3+} + 4OH^- \rightarrow 2AlO(OH) + 2H^+ \tag{1}$$

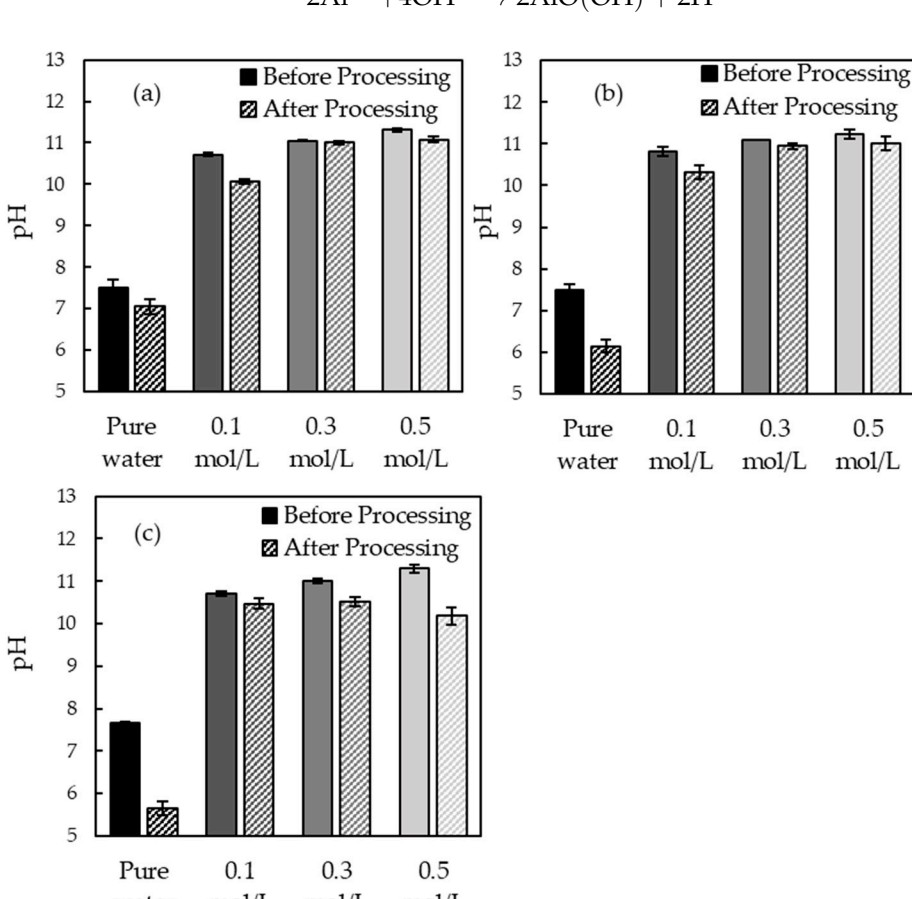

**Figure 1.** pH values before and after steam coating process for: (**a**) 8, (**b**) 16, and (**c**) 24 h.

From Figure 1, it was confirmed that the decreased amount in pH between before and after treatment was changed depending on the concentration of ammonia. The pH after treatment for 24 h decreased significantly in 0.3 and 0.5 mol/L ammonia-treated specimens. The decreased amount in pH was the largest for 0.5 mol/L ammonia-treated specimen. Since the consumption of hydroxy groups by the reaction during the steam coating process increased with increasing concentration, it was likely that the growth of the AlO(OH) film was most enhanced in the 0.5 mol/L ammonia-treated specimen, where the decrease in hydroxyl groups was the highest. Table 2 shows that the standard deviation values were larger after ammonia treatment than before treatment.

**Table 2.** The standard deviation of the pH measurements before and after the treatment.

| Processing Time | Steam Source | Before Processing | After Processing |
|---|---|---|---|
| 8 h | Pure Water | 0.20 | 0.18 |
| | 0.1 mol/L | 0.040 | 0.060 |
| | 0.3 mol/L | 0.010 | 0.045 |
| | 0.5 mol/L | 0.050 | 0.070 |
| 16 h | Pure Water | 0.14 | 0.15 |
| | 0.1 mol/L | 0.11 | 0.16 |
| | 0.3 mol/L | 0.0 | 0.070 |
| | 0.5 mol/L | 0.11 | 0.18 |
| 24 h | Pure Water | 0.050 | 0.16 |
| | 0.1 mol/L | 0.050 | 0.12 |
| | 0.3 mol/L | 0.060 | 0.13 |
| | 0.5 mol/L | 0.10 | 0.20 |

*3.2. Surface Observation*

Figure 2 shows the XRD profiles of the coated films. Several peaks corresponding to aluminum oxide hydroxides, i.e., boehmite ($\gamma$-AlO(OH)), were not observed in the as-received specimens. One set of peaks corresponding to $Mg_2Si$ was observed in all specimens, including the as-received one. These compounds could be derived from the substrate, which formed in the alloy during the steam coating process. $Mg_2Si$ was considered precipitates of Al-Mg-Si alloys, where the precipitation occurred despite the heat energy of steam during the steam coating process. All coated films consisted of aluminum hydroxide, regardless of the treatment duration. No new peaks were generated in the ammonia-added specimens, which indicated that no byproducts were generated due to ammonia addition. The peak intensity of (120) for AlO(OH) at 28.22° was 0.276, 1.51, 2.10, and 5.38 for pure-water and 0.1, 0.3, and 0.5 mol/L for ammonia-treated specimens, respectively. The peak intensity of (120) increased with ammonia concentration. In case of 0.3 mol/L ammonia-treated specimen, the peak intensity of the (020) plane was smaller than that of the other ammonia-treated specimens. Note that the peaks were identified by using JCPDS as Al (PDF card: 01-074-5237), AlO(OH) (PDF card: 00-021-1307), and $Mg_2Si$ (PDF card: 01-077-9648).

The surface images of the substrates subjected to pure water and ammonia treatment (0.1, 0.3, and 0.5 mol/L) for various durations as observed by SEM are shown in Figure 3. It was observed that the crystal morphology of the pure-water-treated specimens was a rectangular crystal, which was identical to the shape of the orthorhombic crystal. In contrast, the crystal morphology of the ammonia-treated specimens was observed as a parallelepiped-shaped crystal, i.e., a rectangular prism with a parallelepiped as its base, which was similar to the shape of the monoclinic crystal. It was also observed that the crystal morphology changed from rectangular to parallelepiped-shaped crystal with the increase in treatment time and concentration. Furthermore, the parallelepiped-shaped crystals were significantly coarsened compared to the rectangular ones. It is known that the surface energy of boehmite crystals changes as the surface state and the crystal form change [35,36]. In this study, the diffraction peak of (120) from XRD analysis was increased, as shown in Figure 2. This suggests that the addition of ammonia in the steam coating process affected the growth of the film.

The percentage, $\gamma$, means the ratio of the number of coarsened parallelepiped-shaped crystals to the number of all crystals on the surface in a straight line on the SEM image:

$$\gamma = h/(h + s) \tag{2}$$

where h and s are the numbers of parallelepiped-shaped crystal and rectangular crystals, respectively.

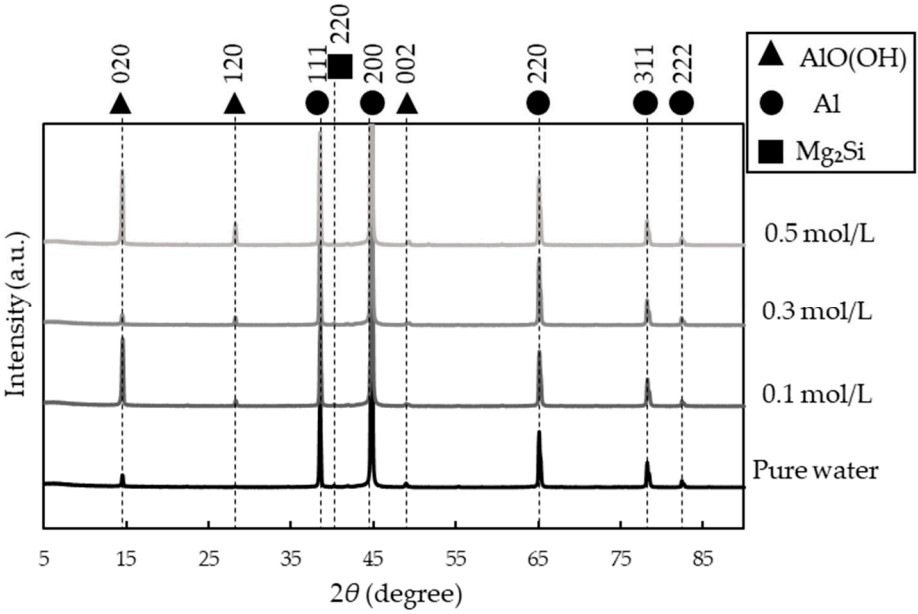

**Figure 2.** XRD profiles of the pure-water-treated specimen and 0.1, 0.3, and 0.5 mol/L ammonia-treated specimens at 200 °C, 24 h.

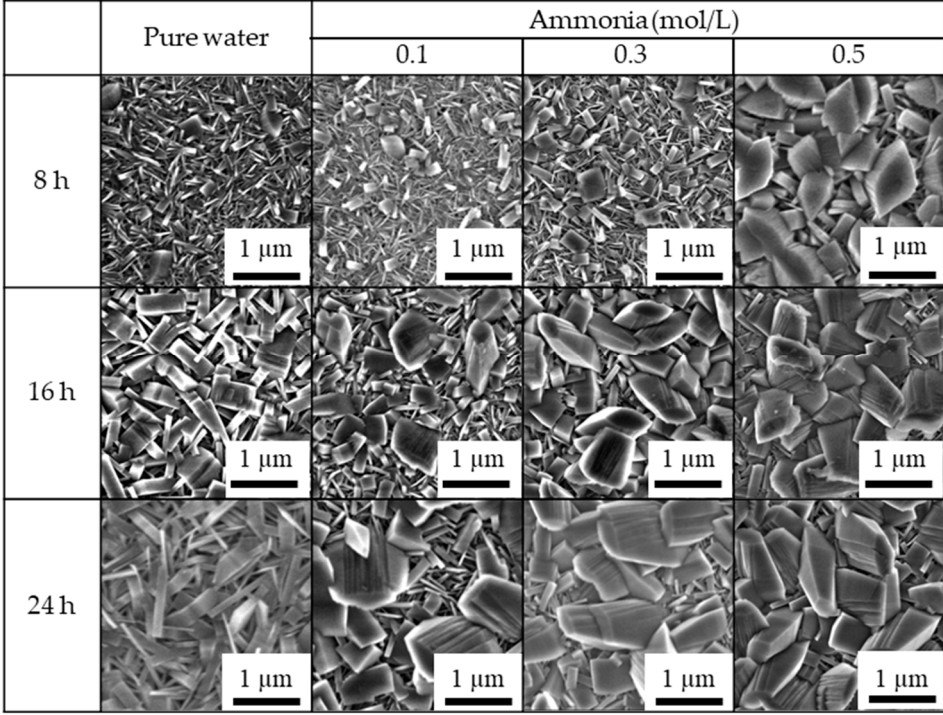

**Figure 3.** SEM images of the surfaces of specimens subjected to steam coating process at 200 °C for various durations: pure-water- and ammonia- (0.1, 0.3, and 0.5 mol/L) treated specimens.

Figure 4 shows the relationship between the total number of crystals (h + s) and γ. The total number of crystals decreased as γ increased. It was observed that γ increased with increasing pH during the treatment. This suggested that there was an inversely proportional relationship between the total number of crystals and γ. The standard deviations of the number of crystal were 1.9, 2.2, 2.8, and 2.1 for pure-water, and 0.1, 0.3, and 0.5 mol/L

for ammonia-treated specimens, respectively. The standard deviations of parallelogram-shaped crystal ratio were 0, 0.037, 0.068, and 0.092 for pure-water, and 0.1, 0.3, and 0.5 mol/L for ammonia-treated specimens, respectively.

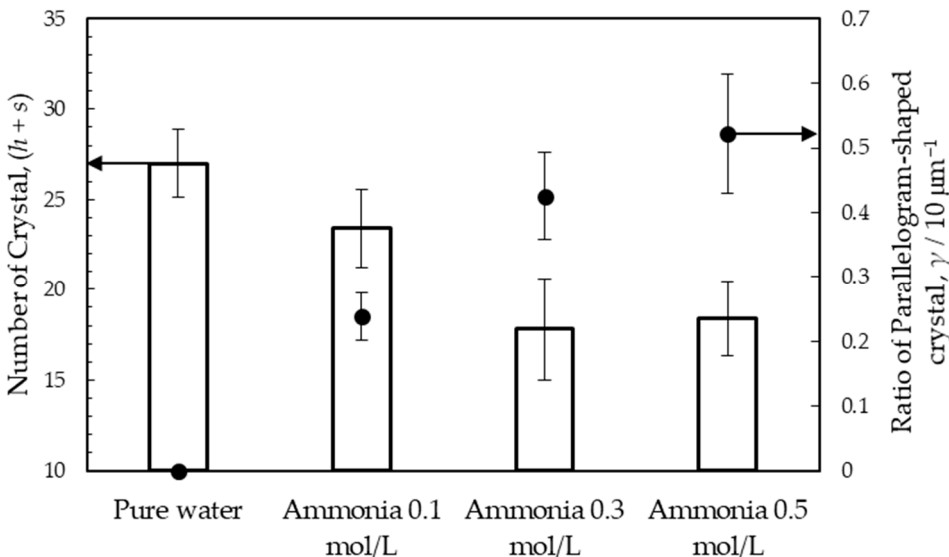

**Figure 4.** Relationship between the total number of crystals and the ratio, $\gamma$ for the pure-water- and ammonia- (0.1, 0.3, and 0.5 mol/L) treated specimens.

*3.3. Growth Rate of AlO(OH) Film Thickness*

The results of the cross-sectional observation after steam coating process at 200 °C using pure water as steam source is shown in Figure 5. In this experiment, the surface was covered with a carbon-derived resin and then the cross-section of specimen was polished using CP. The film thickness was determined from the mapping of elements of Al, O, and C. The change in film thickness with the processing time for each specimen is shown in Figure 6. The standard deviation of film thickness at each time and steam source is shown in Table 3. It was observed that film thickness increased with increasing processing time for all specimens. Film thickness was measured by picking up the five points per specimen, excluding the maximum and minimum values. In the case of pure-water-treated specimens, the film thicknesses for each duration (8, 16, and 24 h) were 0.91, 1.37, and 1.83 µm, respectively. For the 0.1 mol/L ammonia-treated specimens, the film thicknesses for each duration (8, 16, and 24 h) were 1.16, 1.49, and 1.93 µm, respectively. Similarly, for the 0.3 mol/L ammonia-treated specimens, the film thicknesses for each duration (8, 16, and 24 h) were 1.34, 1.56, and 2.16 µm, respectively. For 0.5 mol/L ammonia-treated specimens, the film thicknesses for each duration (8, 16, and 24 h) were 1.39, 1.64, and 2.44 µm, respectively. It was observed that film thickness increased with increasing ammonia concentration at each processing time. By adding ammonia (0.5 mol/L), film thickness increased 1.34 times compared to that of the pure-water-only-treated specimen, when subjected to steam processing at 200 °C for 24 h. The standard deviation was smallest for the pure-water-treated specimen among the specimens treated for the same amount of time, but there was no correlation with the amount of ammonia added.

The growth of the film occurs through diffusion within the base material. In addition, the diffusion of atoms is caused by the movement of ions with a smaller absolute value of the charge [37]. For example, the valency of the Al ion is +3, and that of the hydroxide ion is −1. This suggested that the diffusion of hydroxide ions occurred during the formation of the film, and the formation rate is governed by the diffusion rate of hydroxide ions. Therefore, it is assumed that film growth is caused by the diffusion of hydroxide ion. Thus, the rate of film formation was calculated using Fick's second law, as follows

$$\frac{\partial c}{\partial t} = \frac{\partial}{\partial x}\left(D\frac{\partial c}{\partial x}\right) \tag{3}$$

where c, t, and D are the concentration at position x, processing time, and diffusion coefficient, respectively. For the calculation, ks is used as the time unit (22.4, 57.6, 86.4 ks).

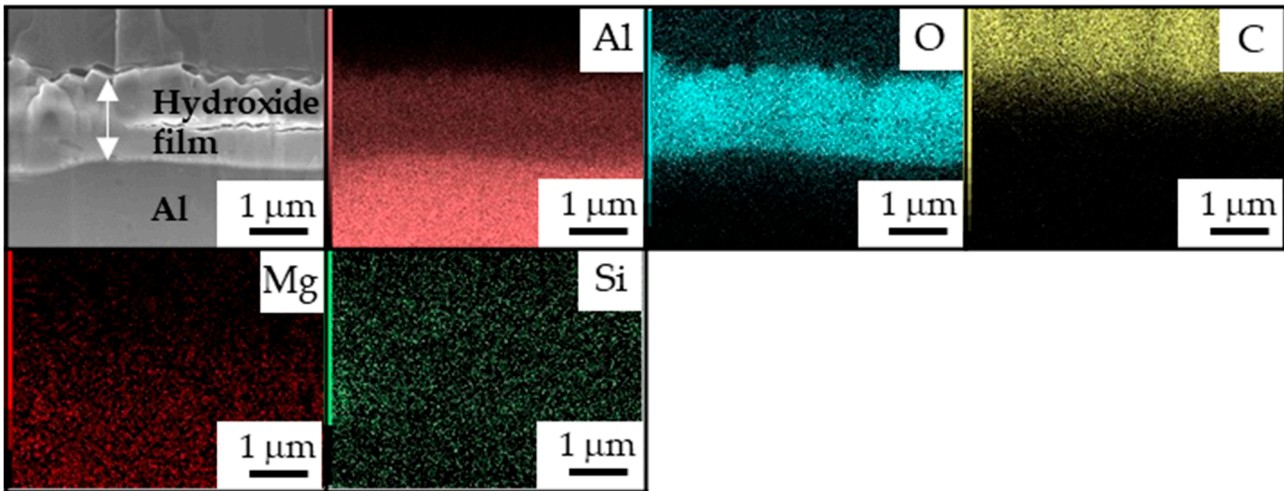

**Figure 5.** The results of cross-sectional observation after steam coating process at 200 °C with pure water as steam source.

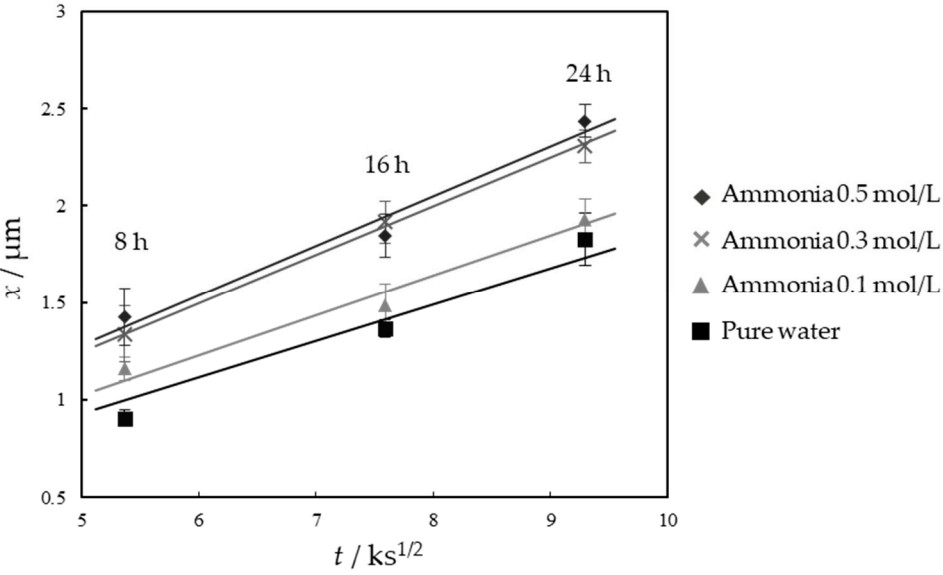

**Figure 6.** Changes in film thickness with the processing time.

**Table 3.** Standard deviation of film thickness at each time and steam source.

| Treatment Time | Pure Water | Ammonia | | |
| --- | --- | --- | --- | --- |
| | | 0.1 mol/L | 0.2 mol/L | 0.5 mol/L |
| 8 h | 0.021 | 0.42 | 0.061 | 0.15 |
| 16 h | 0.095 | 0.43 | 0.11 | 0.11 |
| 24 h | 0.068 | 0.14 | 0.11 | 0.084 |

In this study, the reaction was carried out using saturated steam, where considerable ultrapure water was used in an autoclave. Because the mass transfer rate in the gas phase

was much higher than that in the solid phase, and the addition of ammonia provided an excess of hydroxide ions, the concentration of the hydroxide ions on the surface of the solid phase remained constant, independent of time. Therefore, it was possible to introduce Boltzmann variables from the Boltzmann–Matano method. The equation is thus transformed as follows

$$x = \sqrt{Kt} \tag{4}$$

where K, t, and x are the constant rate of film formation ($\mu m^2/ks$), processing time (ks), and film thickness ($\mu m$), respectively. Figure 7 shows K for pure water and ammonia-treated specimens. Values of K for pure-water and 0.1, 0.3, and 0.5 mol/L ammonia-treated specimens were 0.0363, 0.042, 0.062, and 0.066, respectively. Here, the value of K increased with an increase in the ammonia solution concentration. The value of K for the 0.5 mol/L solution-treated specimen was 1.82 times higher than that of the pure-water-treated specimen. Additionally, the value of K increased sharply from 0.1 to the 0.3 mol/L ammonia solution treatment. This increase was attributed to the saturation of hydroxide ions on the substrate surface between the 0.1 and 0.3 mol/L ammonia-treated specimens. Oversaturation was assumed to have occurred during the 0.5 mol/L ammonia solution treatment.

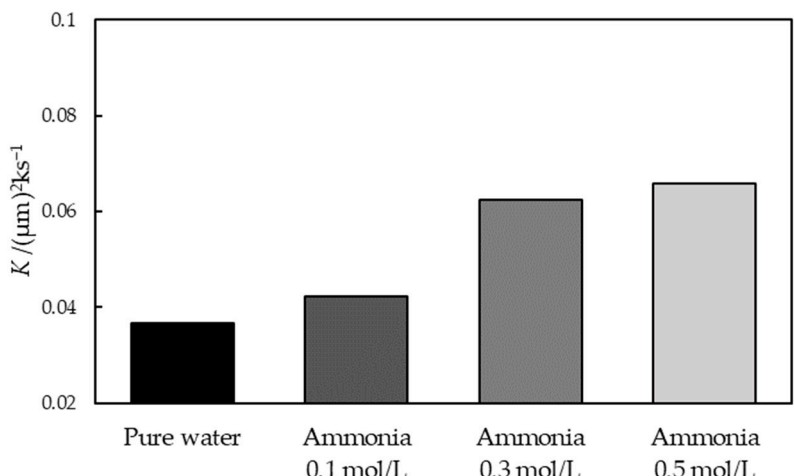

**Figure 7.** Constant rate of the film formation, K, for the pure-water- and ammonia-treated (0.1, 0.3, and 0.5 mol/L) specimens.

## 4. Discussion

### 4.1. Relationship between pH and the Surface Morphology

Figure 3 shows that the surface crystalline morphology changes from rectangular to parallelepiped-shaped crystal as the pH of the solution increases. The value of $\gamma$ increased with increased ammonia concentration, i.e., with an increase in the pH level. This indicates that the increase in pH changed the surface morphology of the crystals and increased the number density of the parallelepiped-shaped crystals. It is known that when crystals are formed and grown from the liquid phase or solution, the solution must be supersaturated. The size of the crystal changes as the driving force increases. The driving force for crystallization depends on the degree of supersaturation of the solution [38]. It is also known that the ionic product increases with increasing temperature [39], and the concentration of the hydroxide ions on the substrate surface increases with an increase in ionic product. In this study, the growth of the film was accelerated by increasing the pH during treatment. This suggests that the crystals did not grow uniformly and showed changes in size due to the accelerated growth of crystals caused by the increase in hydroxide ion concentration on the substrate surface. It is also thought that the crystals grew locally, like dendrite growth, due to the difference in the degree of supersaturation of hydroxide

ions on the substrate surface caused by the increase in hydroxide ion concentration in gas [40].

Figure 8 shows a schematic image of the crystal on the surface during the steam coating process. According to Xin Zhang, boehmite found that the nanoplate morphology can be tuned along specific axes. The growth of boehmite decreased along the [001] direction and increased along the [100] direction as the hydroxide ion concentration increased during the treatment. Based on this discussion, it could be assumed that the [010] direction was the growth direction during the reaction [41]. The lattice constants a, b, and c of AlO(OH) are 2.8681, 12.2336, and 3.6923, respectively, as reported by Charles et al., [42]. In this study, the lattice constants a, b, and c were set to (100), (010), and (001), respectively. However, the intensity of (120) for AlO(OH) was increased by using a higher concentration of ammonia. This phenomenon suggests that ammonia treatment promoted the growth of (120) plane for AlO(OH) or suppressed the growth of (100) and (010) planes. The value of $\gamma$, or the number of coarsened parallelepiped-shaped crystals, increased with increased ammonia concentration. Therefore, the parallelepiped-shaped crystals were formed by significant growth of the Al(OH) crystal. It can be assumed that the surface energy of {120} plane of Al(OH) crystal is lower than that of {100} plane. This hypothesis suggests that the parallelepiped-shaped crystal forms as the total surface energy is reduced when the crystal size grows considerably. On the other hand, the peak intensity of the (020) plane at 0.3 mol/L ammonia-treated specimen was smaller than that of the other ammonia-treated specimens. However, from the trend of the peak intensity in all cases in this study, it can be concluded that (020) and (120) planes compete in the growth of crystals. According to Yaofeng Luo et al., the corrosion behavior of Mg alloys with different crystal planes is different. From this study, it is considered that the crystal planes change due to the coarsened parallelepiped-shaped crystals by ammonia addition. This suggests that the change in corrosion behavior is caused by the change in crystal planes of the crystal due to the ammonia addition [43]. Figure 9 shows a schematic diagram for the facet planes of AlO(OH) crystals of rectangular and parallelepiped-shaped crystals.

### 4.2. Relationship between the Surface Morphology of the Film and the Rate Constant, K

Figure 10 shows the correlation between the rate constant, K, and the number density, $\gamma$, when the steam coating process was applied at 200 °C with ultrapure water and 0.1, 0.3, and 0.5 mol/L of ammonia as a steam source. As K increased, $\gamma$ also increased. This indicated that there is a correlation between $\gamma$ and the film thickness, which in turn increased with an increase in pH. Furthermore, as pH increased, the morphologies of the crystals formed on the crystal surface changed from rectangular to parallelepiped shape, and $\gamma$ increased. In contrast, the growth rate of the obtained film was rate limited due to the diffusion of hydroxide ions. As the concentration of hydroxide ions increased, more concentrated hydroxide ions were present on the substrate surface. These considerations suggested that the K increased with the ammonia concentration. K sharply increased when the ammonia concentration increased from 0.1 to 0.3 mol/L. This might be due to the saturation of the hydroxide ions present on the substrate surface before the process. As shown in Figure 1, the post-treatment pH decreased sharply at the treatment time of 24 h in 0.3 and 0.5 mol/L ammonia-treated specimens compared to the other treatment conditions. This may be due to the fact that hydroxide ions on the substrate surface were saturated in the conditions of 0.3 mol/L and 0.5 mol/L ammonia solution treatment and reacted more actively than in the other conditions. It could explain the increase in K when the ammonia solution concentration increased from 0.3 to 0.5 mol/L. It was thought that K changed from the supersaturated to the unsaturated state during the 0.3 mol/L ammonia solution treatment owing to the increase in the processing time or the reaction time. Here, the relationship between surface morphology and film thickness is discussed. Both the surface morphology of the film and K were presumed to have a causal relationship with pH during the process. However, surface morphology was significantly affected by pH during the initial stage of the crystal growth process, while K was always affected during the

crystal growth process. Therefore, both the surface morphology of the specimens prepared by the steam coating process and the rate constant of the obtained film were affected by a change in pH. Thus, it could be assumed that there was no direct causal relationship between these changes.

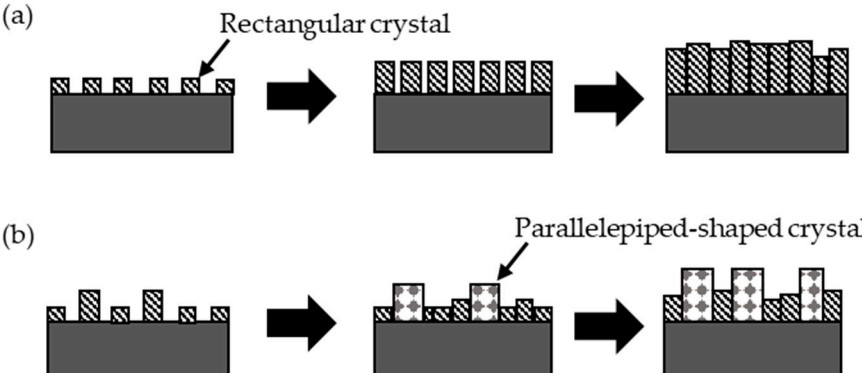

**Figure 8.** A schematic illustration of the changes of surface morphologies and the growth of crystals during the steam coating process. (**a**) Pure-water-treated- and (**b**) ammonia-treated specimens.

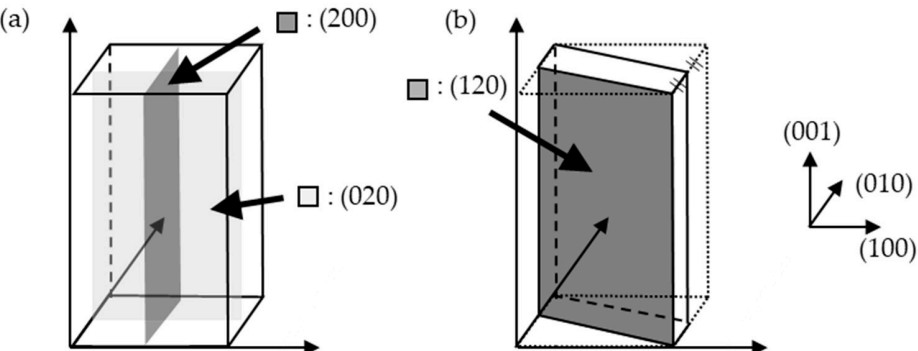

**Figure 9.** A schematic diagram of the facet planes of AlO(OH) crystal corresponding to each diffraction peak in XRD profiles. (**a**) rectangular crystal and (**b**) coarsened parallelepiped-shaped crystal.

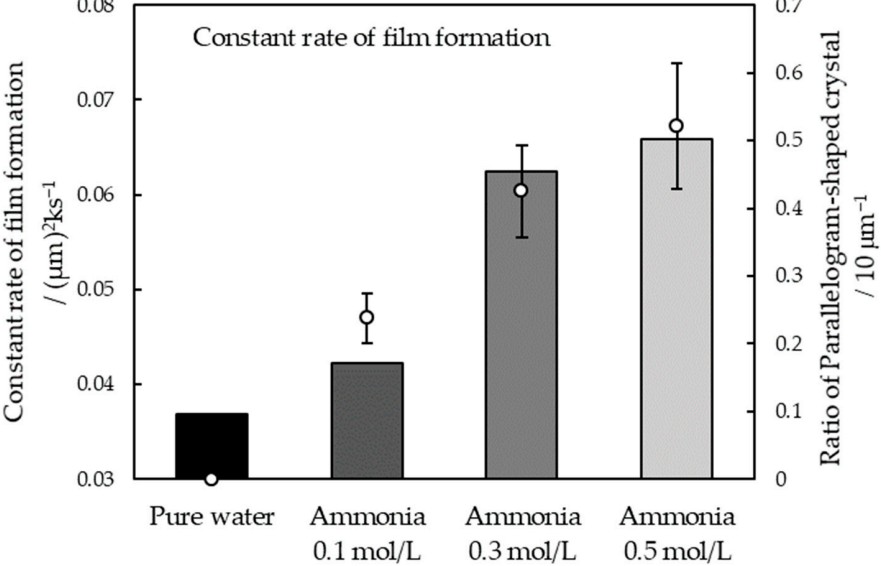

**Figure 10.** Correlation between the rate constant, K, and the ratio of parallelepiped-shaped crystal, $\gamma$, when the steam coating process was performed at 200 °C using pure water and 0.1, 0.3, and 0.5 mol/L of ammonia as a steam source.

## 5. Conclusions

In this study, the effects of ammonia on the growth rate and formation of crystals were successfully demonstrated. The formation rate of the corrosion-resistant film on the Al alloy was increased by adding ammonia to the vapor source during vapor coating on the Al-Mg-Si alloy. Although ammonia was added to the steam source, AlO(OH) could be obtained without byproducts. When ammonia (0.5 mol/L) was added to the vapor coating at 200 °C for 24 h, the thickness of the film increased 1.34 times, and the rate constant increased 1.82 times. It was also confirmed that the added ammonia produced coarsened parallelepiped-shaped crystals on the film surface. Furthermore, as the concentration of ammonia in a steam source increased, the ratio of the parallelepiped-shaped crystal on the surface also increased.

**Author Contributions:** Conceptualization and methodology, A.S.; experimental and data analysis, N.I.; writing—original draft preparation, N.I.; writing—review and editing, A.S. and S.Y.L.; supervision, project administration, and funding acquisition, A.S. All authors have read and agreed to the published version of the manuscript.

**Funding:** This research was supported by the Japan Science and Technology Agency (JST) under the Program on Open Innovation Platform with Enterprises, Research Institute and Academia (OPERA; No. JPMJOP1843) and FOREST Program (No. JPMJFR213N), by Ministry of Economy, Trade and Industry (METI) Monozukuri R&D Support Grant Program for SMEs No. JPJ005698, and by the Japan Society for the Promotion of Science (JSPS) KAKENHI, Grant-in-Aid for Scientific Research (B) (No. 19H02482).

**Institutional Review Board Statement:** Not applicable.

**Informed Consent Statement:** Not applicable.

**Data Availability Statement:** Not applicable.

**Conflicts of Interest:** The authors declare no conflict of interest.

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
