# Peer review of "Effect of Ammonia Addition on the Growth of an AlO(OH) Film during Steam Coating Process"

_coatings, doi:10.3390/coatings12020262_

Round 1

Reviewer 1 Report

the paper deals with a corrosion protection procedure in aluminum alloys. This process uses a stream of water vapor, to which ammonia is added in order to accelerate the process of formation of an anticorrosive patina. The paper demonstrates how it is possible to reduce the times to almost half with the formation of a thicker patina. The characterization techniques employed are appropriate and the methodologies are correct. The paper does not present a high degree of originality but it is certainly of interest to specialists in the sector.
The main problem is the figures. Figure 1 is not present in the text, figure 3 is numbered as figure 1. Figure 2 must be revised as the peak at 220, attributed to Mg2Si is not visible and another peak at 220 is attributed to Al.
The abstract and bibliography are correct. Also the other parts of the text (for example acknowledgments).
I believe that authors need to do a thorough revision of the text which cannot be published in its current form

Author Response

Response to Reviewer 1

Thank you very much for taking the time to review our manuscript. We would like to respond to your useful suggestions and comments as follows (responses are shown in blue, and the results of revisions for each item are also shown). In the revised manuscript, the revised sections are highlighted in yellow color. Thank you very much for your cooperation.

Comments from Reviewer 1:

The paper deals with a corrosion protection procedure in aluminum alloys. This process uses a stream of water vapor, to which ammonia is added in order to accelerate the process of formation of an anticorrosive patina. The paper demonstrates how it is possible to reduce the times to almost half with the formation of a thicker patina. The characterization techniques employed are appropriate and the methodologies are correct. The paper does not present a high degree of originality but it is certainly of interest to specialists in the sector.

The main problem is the figures. Figure 1 is not present in the text, figure 3 is numbered as figure 1. Figure 2 must be revised as the peak at 220, attributed to Mg2Si is not visible and another peak at 220 is attributed to Al.

The abstract and bibliography are correct. Also the other parts of the text (for example acknowledgments).

I believe that authors need to do a thorough revision of the text which cannot be published in its current form.

< Comments from Reviewer 1 >

・Figure 1 is not present in the text

[Answer]

Thank you for pointing this out. We have added the text about Figure 1 and the data that can be read from the pH measurement results. Specifically, we discussed the decrease in pH values before and after treatment, and the significantly the pH values decreased in the case of 0.3 mol/L and 0.5 mol/L ammonia solution treatment for 24 h. The authors thought this phenomenon is related to the rate constant K. We assume that both of these phenomena were caused by the saturation of hydroxide ions on the surface of the sample during treatment.

< Comments from Reviewer 1 >

・Figure 3 is numbered as figure 1.

[Answer]

Thank you for pointing this out. We corrected the figure number.

< Comments from Reviewer 1 >

・Figure 2 must be revised as the peak at 220, attributed to Mg2Si is not visible and another peak at 220 is attributed to Al.

[Answer]

Thank you for pointing out that the peak of Mgâ‚‚Si was different from the one we wanted to show. We modified the figure based on the PDF card (01-077-9648).  

Reviewer 2 Report

REPORTS ON: coatings-1556339

Although the manuscript is reasonable well-organized, the exists some weaknesses suggesting the a MAJOR REVISION be provided, as follow:

  • Firstly, in the Abstract only a simple present tense be used.
  • In section 2, the reproducibility of all experimentations and preparation should be clarified. Besides, error ranges for all physical units should be indicated.
  • For instance, the follow sentence should be replaced : “…results of the
  • 1, 4, 6 and 9 2 should be revised and its corresponding scale bars (error ranges) included.
  • The corresponding JCPDS and/or PDF file number should be included in Fig. 2. Besides, the symbols should be rewritten inside Figure.
  • Although the thicknesses of films are indicated in Fig. 5, it is not clearly evidenced each one of these thicknesses. At least a micrograph should be included in order to depict these measurements or their average values (an representative figure).
  • Since Fig. 6 is plotted based on those results of the film thicknesses, their corresponding error ranges should be determined. It seems that determining the error ranges, both the pure and 0.1 ammonia samples have similar results. Similarly, the 0.3 and 0.5 seem to depict equivalent results. This is a reason that the error ranges should be included.
  • When analyzing the list of references, only of about 4% of the list of references are after 2018th. This indicates that an up-to-date should be provided. Additionally, few articles considering COATING journal is cited. This suggested that when up-to-dating the manuscript, some references recently published in Coating should be mentioned.

Author Response

Response to Reviewer 2

Thank you very much for taking the time to review our manuscript. We would like to respond to your useful suggestions and comments as follows (responses are shown in blue, and the results of revisions for each item are also shown). In the revised manuscript, the revised sections are highlighted in yellow color. Thank you very much for your cooperation.

Comments from Reviewer 2:

Although the manuscript is reasonable well-organized, the exists some weaknesses suggesting the a MAJOR REVISION be provided, as follow:

Firstly, in the Abstract only a simple present tense be used.

In section 2, the reproducibility of all experimentations and preparation should be clarified. Besides, error ranges for all physical units should be indicated.

For instance, the follow sentence should be replaced : “…results of the

1, 4, 6 and 9 2 should be revised and its corresponding scale bars (error ranges) included.

The corresponding JCPDS and/or PDF file number should be included in Fig. 2. Besides, the symbols should be rewritten inside Figure.

Although the thicknesses of films are indicated in Fig. 5, it is not clearly evidenced each one of these thicknesses. At least a micrograph should be included in order to depict these measurements or their average values (an representative figure).

Since Fig. 6 is plotted based on those results of the film thicknesses, their corresponding error ranges should be determined. It seems that determining the error ranges, both the pure and 0.1 ammonia samples have similar results. Similarly, the 0.3 and 0.5 seem to depict equivalent results. This is a reason that the error ranges should be included.

When analyzing the list of references, only of about 4% of the list of references are after 2018th. This indicates that an up-to-date should be provided. Additionally, few articles considering COATING journal is cited. This suggested that when up-to-dating the manuscript, some references recently published in Coating should be mentioned.

< Comments from Reviewer 2>

・Firstly, in the Abstract only a simple present tense be used.

[Answer]

Thank you for pointing this out. Our authors changed the introduction part with present tense as your comment.

 < Comments from Reviewer 2>

・In section 2, the reproducibility of all experimentations and preparation should be clarified. Besides, error ranges for all physical units should be indicated. For instance, the follow sentence should be replaced : “…results of the 1, 4, 6 and 9 2 should be revised and its corresponding scale bars (error ranges) included.

[Answer]

Thank you for pointing this out. This experiment was tested in duplicate, and our authors added this information in the text. As follow your comment, our authors also inputted a standard deviation to all the data to clarify. For the graph of the rate constant K in Figure 7, we could not indicate the standard deviation because it was created based on the average value of each condition.

 < Comments from Reviewer 2>

・The corresponding JCPDS and/or PDF file number should be included in Fig. 2. Besides, the symbols should be rewritten inside Figure.

[Answer]

Thank you for the comment. Our authors added the information about the corresponding PDF file number used to identify all peaks in the XRD results; Al (PDF card: 01-074-5237), AlO(OH) (PDS card: 00-021-1307), and Mgâ‚‚Si (PDF card: 01-077-9648).

 < Comments from Reviewer 2>

・Although the thicknesses of films are indicated in Fig. 5, it is not clearly evidenced each one of these thicknesses. At least a micrograph should be included in order to depict these measurements or their average values (an representative figure).

[Answer]

Thank you for pointing this out. In this experiment, we selected five points to measure the film thickness and then calculated the average value with three points, excluding the maximum and minimum values, respectively. We added the specific film thickness values in the text. In addition, we added a micrograph of a specific cross-section treated sample which was treated with pure water as the steam source at 200°C for 24 h.

 < Comments from Reviewer 2>

Since Fig. 6 is plotted based on those results of the film thicknesses, their corresponding error ranges should be determined. It seems that determining the error ranges, both the pure and 0.1 ammonia samples have similar results. Similarly, the 0.3 and 0.5 seem to depict equivalent results. This is a reason that the error ranges should be included.

[Answer]

Thank you for pointing this out. The authors added the information of a standard deviation of the film thickness to the figure and the text as Table 3. As the reviewer pointed out, the difference in the rate constant K was relatively small between pure water sample and 0.1 mol/L of ammonia solution treated sample in Figure 6, however a standard deviation was also small enough. On the other hand, 0.3 mol/L sample and 0.5 mol/L sample seemed to depict equivalent results as the reviewer pointed out, however it was demonstrated that the film thickness was more significant in 0.5 mol/L sample due to the shifted range of the standard deviation.

 < Comments from Reviewer 2>

When analyzing the list of references, only of about 4% of the list of references are after 2018th. This indicates that an up-to-date should be provided. Additionally, few articles considering COATING journal is cited. This suggested that when up-to-dating the manuscript, some references recently published in Coating should be mentioned.

[Answer]

Thank you for your suggestion. Our authors added 10 more papers which were published after 2018. Especially, 5 papers from Coating are cited this time.

Reviewer 3 Report

In this manuscript, by means of adding ammonia during the boiling step in the steam process, the author succeeded in increasing the film formation rate during steam coating process. Furthermore, AlO(OH) crystals were successfully synthesized without byproducts, even when ammonia was added.

But as far as I am concerned, it still has some small problems.

  1. In this manuscript, it has said that adding ammonia will decrease the environment pollution, but the ammonia also can cause damage to our health.
  2. In Figure 1 (a) and (b), in the 0.5mol/L ammonia solution treatment, the decrease in hydroxyl groups is of a small amount, even less than the 0.1mol/L ammonia solution treatment, but in Figure 1 (c), the amount of hydroxyl groups dropped off suddenly. However, the author just said generally that the decrease in hydroxyl groups is the highest in the 0.5mol/L ammonia solution treatment.
  3. In Figure 2, the cutline does not say clearly that how long had the tested sample been boiled.
  4. In Figure 2, the peak intensity of (020) in the 0.3mol/L ammonia solution treatment dropped sharply compared to 0.1mol/L. However, only the change of the peak intensity of (120) is expounded by the author.
  5. At line 50, the cutline is wrong, it should be Figure 3, not Figure 1.
  6. In Figure 4 and Figure 9, the unit of γ is different.

Author Response

Response to Reviewer 3

Thank you very much for taking the time to review our manuscript. We would like to respond to your useful suggestions and comments as follows (responses are shown in blue, and the results of revisions for each item are also shown). In the revised manuscript, the revised sections are highlighted in yellow color. Thank you very much for your cooperation.

Comments from Reviewer 3:

In this manuscript, by means of adding ammonia during the boiling step in the steam process, the author succeeded in increasing the film formation rate during steam coating process. Furthermore, AlO(OH) crystals were successfully synthesized without byproducts, even when ammonia was added.

But as far as I am concerned, it still has some small problems.

In this manuscript, it has said that adding ammonia will decrease the environment pollution, but the ammonia also can cause damage to our health.

In Figure 1 (a) and (b), in the 0.5mol/L ammonia solution treatment, the decrease in hydroxyl groups is of a small amount, even less than the 0.1mol/L ammonia solution treatment, but in Figure 1 (c), the amount of hydroxyl groups dropped off suddenly. However, the author just said generally that the decrease in hydroxyl groups is the highest in the 0.5mol/L ammonia solution treatment.

In Figure 2, the cutline does not say clearly that how long had the tested sample been boiled.

In Figure 2, the peak intensity of (020) in the 0.3mol/L ammonia solution treatment dropped sharply compared to 0.1mol/L. However, only the change of the peak intensity of (120) is expounded by the author.

At line 50, the cutline is wrong, it should be Figure 3, not Figure 1.

In Figure 4 and Figure 9, the unit of γ is different.

 < Comments from Reviewer 3>

・In this manuscript, it has said that adding ammonia will decrease the environment pollution, but the ammonia also can cause damage to our health.

[Answer]

Thank you for the comment. During this experiment, liquid waste was generated by ammonia treatment containing aluminum. Therefore, it is a big issue about the aluminum separation from the generated liquid waste. Currently, the development of denitrification technology for ammonia wastewater treatment has been progressing [1-2]. In this contribution, we considered that ammonia is an alkaline solution with low environmental impact. Ammonia stripping method is used to treat the ammonia wastewater, and it is reported that ammonia release rate and removal rate increase when the pH values are getting higher [3]. In this experiment, the pH value of the solution after treatment was more than 10, which is a suitable condition for using the ammonia stripping method. In addition, there is some research to apply for ammonia recovering in fuel cells. From the viewpoint of recycling, ammonia is considered to have a low environmental impact as well [4]. From these backgrounds, the authors mentioned that this methodology is an environmentally friendly synthetic route in this paper.

References

  1. L, Manasa.; Alka, M. Current perspectives of anoxic ammonia removal and blending of partial nitrifying and denitrifying bacteria for ammonia reduction in wastewater treatment. J. Water Process. Eng. 2021, 41, 1-12.
  2. Qianru, W.; Jianping. G.; Ping, C. The impact of alkali and alkaline earth metals on green ammonia synthesis. Chem. 2021, 7, 3203-3220.
  3. Atushi, W. Innovative System for Nitrogen Removal of Industrial Waste Water. Surf. Finish. Soc. Jpn. 1997, 48, 257-260.
  4. Shogo, O.; Takashi, S.; Effective utilization of the recovery ammonia from industrial waste water. JIE. 2016, P-09, 288-289.

< Comments from Reviewer 3>

・In Figure 1 (a) and (b), in the 0.5mol/L ammonia solution treatment, the decrease in hydroxyl groups is of a small amount, even less than the 0.1mol/L ammonia solution treatment, but in Figure 1 (c), the amount of hydroxyl groups dropped off suddenly. However, the author just said generally that the decrease in hydroxyl groups is the highest in the 0.5mol/L ammonia solution treatment.

[Answer]

Thank you for the comment. We added the result of pH value after ammonia treatment, and the result revealed that the pH values decreased significantly in 0.3 mol/L and 0.5 mol/L ammonia solution treatment after 24 h. The authors corrected the text so the context does not mislead readers.

< Comments from Reviewer 3>

・In Figure 2, the cutline does not say clearly that how long had the tested sample been boiled.

[Answer]

Thank you for the comment. We added the information of the treatment conditions (200 °C, 24 h) to the title of the figure.

< Comments from Reviewer 3>

・In Figure 2, the peak intensity of (020) in the 0.3mol/L ammonia solution treatment dropped sharply compared to 0.1mol/L. However, only the change of the peak intensity of (120) is expounded by the author.

[Answer]

Thank you for your comment. As you mentioned, the peak intensity of the (020) plane at 0.3 mol/L ammonia solution treated specimen was smaller than that of the other ammonia treatment. However, when we read the trend of the peak intensity in all cases in this study, it can be claimed that (020) and (120) planes compete in the growth of crystals. Of course, this phenomenon has not yet been clarified, so we agree that further study is needed.

< Comments from Reviewer 3>

・At line 50, the cutline is wrong, it should be Figure 3, not Figure 1.

[Answer]

Thank you for pointing this out. The authors corrected it.

< Comments from Reviewer 3>

・In Figure 4 and Figure 9, the unit of γ is different.

[Answer]

Thanks for pointing that out, the authors revised it from mm to mm.

Round 2

Reviewer 2 Report

Based on the revised version of this proposed manuscript, in my frank view, it deserves its final publication.

Best Regards;

Reviewer 3 Report

The authors have addressed my comments. It is in a good shape to publish now.